# Generative Occupancy Fields for 3D Surface-Aware Image Synthesis

**Xudong Xu**[†]     **Xingang Pan**[‡]     **Dahua Lin**[†]     **Bo Dai**[§]
[†]CUHK - SenseTime Joint Lab, The Chinese University of Hong Kong
[‡]Max Planck Institute for Informatics  [§]S - Lab, Nanyang Technological University
[†]{xx018, dhlin}@ie.cuhk.edu.hk  [‡]xpan@mpi-inf.mpg.de  [§]bo.dai@ntu.edu.sg

## Abstract

The advent of generative radiance fields has significantly promoted the development of 3D-aware image synthesis. The cumulative rendering process in radiance fields makes training these generative models much easier since gradients are distributed over the entire volume, but leads to diffused object surfaces. In the meantime, compared to radiance fields occupancy representations could inherently ensure deterministic surfaces. However, if we directly apply occupancy representations to generative models, during training they will only receive sparse gradients located on object surfaces and eventually suffer from the convergence problem. In this paper, we propose **Generative Occupancy Fields (GOF)**, a novel model based on generative radiance fields that can learn compact object surfaces without impeding its training convergence. The key insight of GOF is a dedicated transition from the cumulative rendering in radiance fields to rendering with only the surface points as the learned surface gets more and more accurate. In this way, GOF combines the merits of two representations in a unified framework. In practice, the training-time transition of *start from radiance fields and march to occupancy representations* is achieved in GOF by gradually shrinking the sampling region in its rendering process from the entire volume to a minimal neighboring region around the surface. Through comprehensive experiments on multiple datasets, we demonstrate that GOF can synthesize high-quality images with 3D consistency and simultaneously learn compact and smooth object surfaces. Our code is available at https://github.com/SheldonTsui/GOF_NeurIPS2021.

## 1 Introduction

Deep generative adversarial networks [1–4] have demonstrated their superiority in synthesizing photorealistic and striking images. However, these models are often constrained in the 2D domain, struggling to generate 3D consistent images, let alone grasping the underlying 3D object shapes. 3D-aware image synthesis thus becomes an appealing and promising choice as it learns a 3D representation explicitly from a collection of unposed images. Consequently, it can not only synthesize 3D consistent images by manually controlling the rendering camera poses, but also pave the way for various downstream tasks such as shape editing and relighting.

Inspired by the success of neural radiance fields (NeRF) [5] in 3D scene modeling, recent 3D-aware generative models, referred to as generative radiance fields (GRAFs), have applied NeRF as the explicit 3D representation for image synthesis [6, 7]. With the help of NeRF, they are capable of hallucinating photorealistic images in a 3D consistent manner. Moreover, since NeRF holds the superior ability for rendering translucent objects by compositing colored densities along each ray in its volume rendering process, it also significantly facilitates the training of GRAFs as gradients are naturally distributed over the entire volume. However, they still incur an inevitable incapacity of

35th Conference on Neural Information Processing Systems (NeurIPS 2021).

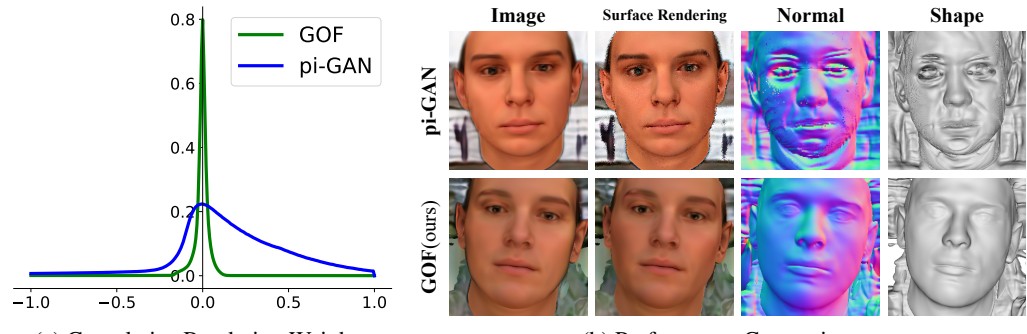

(a) Cumulative Rendering Weights      (b) Performance Comparison

Figure 1: **(a)** The cumulative rendering weights (color weights) of our approach GOF more focus on the surface (y-axis) than previous methods like pi-GAN [6], which indicates our predicted volume densities more concentrate on the object surfaces. **(b)** Owing to the diffused volume densities, the preceding method pi-GAN captures messy surface normals and object shapes. Moreover, the image rendered only with the surface points is quite noisy. In contrast, more surface-centralized densities predicted by our method ensure compact and smooth object surfaces thus enable a high-quality surface rendering during inference. (**Zoom in for best view**)

capturing an accurate and compact object surface. As shown in Fig. 1(a), the state-of-the-art GRAF model pi-GAN is prone to predict diffused object surfaces, as the volume densities are smoothly spread around the surfaces. Such diffused surfaces could significantly hamper the applications of GRAF models in downstream tasks such as shape recovery. Moreover, under different light conditions, the artifacts of surfaces could be amplified and inherited through the rendering process, resulting in synthesized images that are messy and faulty.

In this work, we propose **Generative Occupancy Fields (GOF)**, a novel GRAF-like image synthesis model that can learn compact object surfaces. GOF is inspired by the design of occupancy networks [8] that implicitly represents a 3D surface with the continuous decision boundary of a neural classifier. In this way, occupancy networks are capable of effectively locating surfaces via root-finding and encouraging the compactness of modeled surfaces inherently. However, GOF avoids directly applying such a design to 3D-aware image synthesis. While occupancy networks require precise object masks to train [9, 10], a more crucial factor is that they rely on the surface points for differentiable rendering [11, 12, 9, 13]. A generative model equipped with occupancy representations will thus meet severe convergence problems during training due to the sparsity of gradients. To unify the merits of both NeRF and occupancy networks for 3D-aware image synthesis, GOF adopts the design of GRAFs and at the same time leverages a nontrivial transition from the cumulative rendering to rendering with only the surface points, *i.e. start from radiance fields and march to occupancy representations*. Specifically, GOF will reinterpret the alpha values in the cumulative rendering process as occupancy values, so that it can locate the learned surface via root-finding. Subsequently, it can naturally encourage the compactness of learned surfaces by gradually shrinking the sampling region in the rendering process from the entire volume to a minimal neighboring region around the surface.

Thanks to the unified integration of radiance fields and occupancy representations, GOF can benefit from the representation effectiveness of radiance fields while ensuring the compactness of learned object surfaces through the shrinking process. As presented in Fig. 1(a), in GOF the distribution of cumulative rendering weights concentrates more closely around object surfaces compared to that of pi-GAN, eventually resulting in a compact and smooth surface. Moreover, GOF can thus alternatively render an image only with points on the learned surfaces like occupancy networks as illustrated in Fig. 1 (b). And during inference such a rendering scheme has the potential to alleviate the burden of sampling a large number of points along each ray for synthesizing a single image. Through exhaustive experiments on synthetic and real-world datasets, we demonstrate that GOF can achieve state-of-the-art performance on 3D-aware image synthesis. Meanwhile, it is capable of capturing compact and accurate 3D shapes that empower its applications in various downstream tasks such as 3D shape reconstruction. We validate this point by quantitative results of 3D shape reconstruction on the Synface dataset. Finally, we have also verified the ability of GOF in rendering high-quality images with only the surface points, which is hardly achievable in previous approaches.

## 2 Related Work

**Neural implicit function for 3D representations.** A plethora of works [14, 8, 15–21] has exploited neural implicit functions for 3D geometry modeling. Among these works, neural radiance fields (NeRF) [5] has attracted growing attention due to its compelling results on novel view synthesis. It leverages an MLP network to approximate the radiance fields of static 3D scenes. And by learning to reconstruct existing views, it is capable of capturing 3D geometric details from only 2D supervision. A series of succeeding variants of NeRF have been proposed to improve it, including utilizing the spatial sparsity to reduce its computational complexity [22, 23], refining the rendering process to improve its efficiency [24, 25], as well as adopting reflectance decompositions to enhance its modeling capacity [26, 27]. There are also works that capitalize on the differentiable rendering of neural implicit functions for 3D reconstruction [11, 12, 9, 13, 28, 29]. Specifically, SDFDiff [12] relies on the interpolation of eight neighboring SDF samples around the surface intersection to obtain the derivatives, while Atzmon *et al* [28] use a sample network to relate samples' positions to network parameters and thus achieve an improved generalization ability. More interestingly, by adopting occupancy representations, DVR [11] and IDR [9] show volumetric rendering is inherently differentiable so that network parameters can be optimized directly with derived analytic gradients. Different from methods aforementioned above, GOF is a generative model for 3D-aware image synthesis that can learn 3D representations from a set of 2D images with unknown camera poses.

**Generative 3D-aware image synthesis.** In order to synthesize 3D consistent images, researchers have explored a lot on how to incorporate 3D representations into the classical GAN model [1]. Some methods [30–32] resort to learning from 3D data directly, yet the requirement of 3D supervision limits their practical applicability. A more appealing alternative is thus learn from unposed 2D images in an unsupervised manner. Preceding works along this line of research adopt voxels as their intermediate 3D representations [33–35] and achieve explicit control over the pose of synthesized images. Inspired by the superior representation capacity of radiance fields over voxels, recent attempts [6, 7, 36] have replaced voxels with neural radiance fields [5] to improve the fidelity of synthesized 3D consistent images. Despite the striking performance, these models, referred to as generative radiance fields (GRAFs), tend to predict diffused object surfaces, which impedes its applicability in various downstream tasks. In this work, GOF aims at resolving this problem of GRAFs by combining them with the perspective of occupancy networks [8] and recent successes of recovering smooth and accurate shapes from natural images [37–40]. Recently, three concurrent works, UNISURF [10], NeuS [41] and VolSDF [42], also combine implicit surfaces and radiance fields in a unified framework, sharing similar spirits with our proposed GOF but different in tasks and focuses. Specifically, They focus on multi-view 3D reconstruction and attempt to alleviate the requirement of training-time precise masks through the integration of radiance fields and occupancy representations. Nevertheless, they still require images with ground-truth poses for training. By contrast, GOF targets on the challenging task of 3D-aware image synthesis, where the synthesized images should be not only natural and vivid, but also consistent in the 3D space. By integrating radiance fields and occupancy representations, GOF is able to facilitate the convergence of GRAFs and ensure the compactness of learned object surfaces. Compared to existing GRAFs, the applicability of GOF is thus significantly broadened.

## 3 Methodology

We propose **generative occupancy fields (GOF)**, a novel synthesis model, belonging to generative radiance fields and aiming to learn from unposed images. Conditioned on a latent code $\mathbf{z} \sim p_{\mathbf{z}}$, our generator $g_\theta$ can generate a 3D radiance field $\mathbf{R}$, from which we can render a realistic image with a sampled camera pose $\xi \sim p_\xi$ and simultaneously recover smooth and compact object surfaces. In the following, we first present the background of neural radiance fields, and then introduce our proposed GOF model in detail.

### 3.1 Neural Radiance Fields

We adopt neural radiance fields (NeRF) as our explicit 3D representation for image synthesis, owing to its strong performance in novel view synthesis on complex scenes. NeRF represents a static scene as per-point volume densities and view-dependent RGB colors. Given a 3D point $\mathbf{x} \in \mathbb{R}^3$ in space and a view direction $\mathbf{d} \in \mathbb{R}^3$, NeRF capitalizes on a multi-layer perceptron (MLP) to predict the

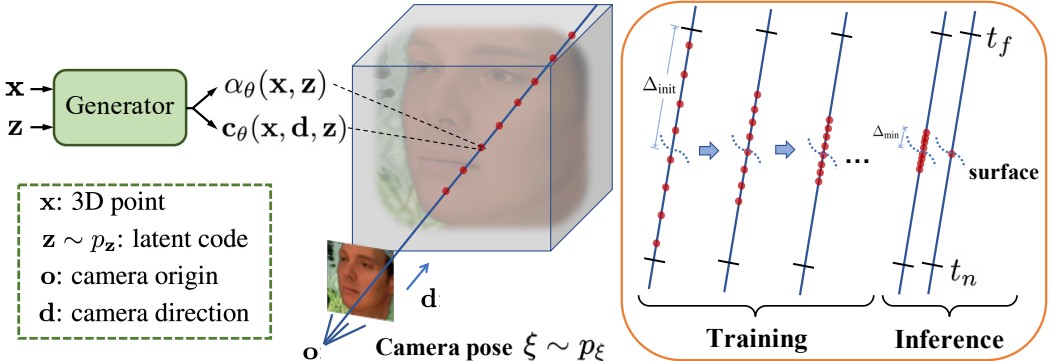

Figure 2: **Shrinking process**. During the training, the sampling interval $\Delta$ is initially a half of the distance between near $t_n$ and far bounds $t_f$ and shrinks gradually to a pre-defined value $\Delta_{\text{min}}$. For inference, we can use cumulative rendering by sampling points in the minimal interval $\Delta_{\text{min}}$ and alternatively render only with the surface points.

volume density $\sigma(\mathbf{x}) \in \mathbb{R}$ and the emitted color $\mathbf{c}(\mathbf{x}, \mathbf{d}) \in \mathbb{R}^3$. To render a novel view for the scene, NeRF leverages the classic volume rendering technique [43] to estimate the color of each pixel. It starts by accumulating the colored densities of $N$ points $\{\mathbf{x}_i = \mathbf{o} + t_i\mathbf{d}\}$ sampled within near and far bounds $[t_n, t_f]$ along the camera ray $\mathbf{r}(t) = \mathbf{o} + t\mathbf{d}$, where $\mathbf{o}$ stands for the camera origin. The integrated color is then estimated via alpha composition as follows:

$$\hat{\mathbf{C}}(\mathbf{r}) = \sum_{i=1}^{N} T_i \Big(1 - \exp\big(-\sigma(\mathbf{x}_i)\delta_i\big)\Big)\mathbf{c}(\mathbf{x}_i, \mathbf{d}), \text{ where } T_i = \exp(-\sum_{j=1}^{i-1} \sigma(\mathbf{x}_j)\delta_j), \quad (1)$$

where $\delta_i = |x_{i+1} - x_i|$ is the distance between adjacent points. Note that, equation (1) is naturally differentiable and NeRF can be directly optimized through the reconstruction error of existing views.

### 3.2 3D Surface-Aware Image Synthesis via Generative Occupancy Fields

To apply NeRF as the 3D representation, the proposed generative occupancy fields (GOF) incorporates an additional latent code $\mathbf{z} \sim p_{\mathbf{z}}$ into NeRF, such that synthesizing an image follows a reformulated cumulative rendering process:

$$\hat{\mathbf{C}}(\mathbf{r}, \mathbf{z}) = \sum_{i=1}^{N} T_i \Big(1 - \exp\big(-\sigma_\theta(\mathbf{x}_i, \mathbf{z})\delta_i\big)\Big)\mathbf{c}_\theta(\mathbf{x}_i, \mathbf{d}, \mathbf{z}), \text{ where } T_i = \exp(-\sum_{j=1}^{i-1} \sigma_\theta(\mathbf{x}_j, \mathbf{z})\delta_j). \quad (2)$$

However, directly training GOF according to Eq.(2) fails to maintain the surface compactness as reported in previous approaches [6, 7]. Actually, such a defect arises from an inevitable "shape-color ambiguity" of the cumulative rendering process, *i.e.*, small perturbations on surfaces still lead to realistic RGB images which are enough to fool the discriminator.

Owing to the constrained range of poses seen at training, the discriminator is less motivated to further concentrate the color weights $w_i = T_i\big(1 - \exp(-\sigma_\theta(\mathbf{x}_i, \mathbf{z})\delta_i)\big)$ aforementioned in Fig. 1 (a) on the exact object surface. On the other hand, we observe that although leading to diffused surfaces at the end, color weights $w_i$ gradually concentrate around the object surface as the training proceeds. Inspired by this observation, in GOF we propose a training-time operation to facilitate the concentration of color weights $w_i$. The basic idea is gradually shrinking the sample region in the cumulative rendering process from the entire volume to a narrow interval around the surface, so that color weights are enforced to continuously move towards the exact surface.

To enable the proposed training-time shrinking process, GOF is required to locate the surface by thresholding the predicted densities $\sigma_\theta(\mathbf{x}, \mathbf{z})$, assuming points on the surface have the largest densities, However, values of the densities predicted in generative radiance fields could range from 0 to 50, making it hard to determine an effective threshold $\tau$ during the whole training period. On the other

hand, in the cumulative rendering process shown in Eq.(2), we found that *the intermediate alpha values* used for numerical stability inherently fall in a fixed value range as:

$$\alpha_\theta(\mathbf{x}_i, \mathbf{z}) = 1 - \exp(-\sigma_\theta(\mathbf{x}_i, \mathbf{z})\delta_i) \in [0, 1]. \tag{3}$$

More importantly, these alpha values $\alpha_\theta(\mathbf{x}, \mathbf{z})$ come close to 1 for points in the occupied space while approaching 0 for points in the free space, making them resemble the occupancy values [8] in both quantity and semantics. Inspired by the similarity, we thus propose to reformulate generative radiance fields by predicting alpha values $\alpha_\theta(\mathbf{x}, \mathbf{z})$ directly instead of volume densities $\sigma_\theta(\mathbf{x}, \mathbf{z})$. In the mean time, we reinterpret the alpha values as occupancy values, and subsequently locate surfaces with root-finding, a more effective strategy originated in occupancy networks [11]. According to the above reformulation and reinterpretation, we thus dub our method as **Generative Occupancy Fields**.

As GOF estimates alpha values instead of volume densities, the original volume rendering process in Eq.(2) conditioned on the latent code $\mathbf{z}$ is reformulated as

$$\hat{\mathbf{C}}(\mathbf{r}, \mathbf{z}) = \sum_{i=1}^{N} \alpha_\theta(\mathbf{x}_i, \mathbf{z}) \prod_{j<i} \big(1 - \alpha_\theta(\mathbf{x}_j, \mathbf{z})\big) \mathbf{c}_\theta(\mathbf{x}_i, \mathbf{d}, \mathbf{z}), \tag{4}$$

where the value range of $\alpha_\theta(\mathbf{x}_i, \mathbf{z})$ is guaranteed with a sigmoid function. And to locate the surface via root-finding, for a specific ray $\mathbf{r}(t) = \mathbf{o} + t\mathbf{d}$ we will evenly sample $M$ points $\{\mathbf{x}_k = \mathbf{o} + t_k\mathbf{d}; k = 1, ..., M\}$ that partition the entire volume $[t_n, t_f]$ into $M$ equally-spaced bins. After obtaining the corresponding alpha values $\{\alpha_\theta(\mathbf{x}_k, \mathbf{z}); k = 1, ..., M\}$ by querying the generator $g_\theta$, the surface $\mathcal{S}$ is located in the $k^{\mathcal{S}}$-th bin where $\alpha_\theta$ changes *for the first time* from free space ($\alpha_\theta < \tau$) to occupied space ($\alpha_\theta < \tau$):

$$k^{\mathcal{S}} = \underset{k}{\operatorname{argmin}} \big(\alpha_\theta(\mathbf{x}_k, \mathbf{z}) < \tau \le \alpha_\theta(\mathbf{x}_{k+1}, \mathbf{z})\big), \tag{5}$$

where $\tau$ is a pre-defined threshold. In practice, we empirically set $\tau$ as 0.5. In order to find the surface point $\mathbf{x}_s = \mathbf{o} + t_s\mathbf{d}$ more precisely, we further apply the above secant method iteratively for $m_s$ times, resulting in a fine-grained bin $[\mathbf{x}_{k^{\mathcal{S}}}, \mathbf{x}_{k^{\mathcal{S}}+1}]$. It's worth noting that the $M$ sampled points are only used for root-finding. Thus they do not require the computation of gradients in the implementation.

Based on the located surface $\mathcal{S}$, we can thus successfully conduct the proposed shrinking process, which is schematically elaborated in Fig. 2. Specifically, when sampling $N$ points for Eq.(4) at each training step, we will only sample within a region neighboring the surface $[t_s - \Delta, t_s + \Delta]$:

$$t_i \sim \mathcal{U}\left[t_s - \Delta + \frac{2i - 2}{N}\Delta, t_s - \Delta + \frac{2i}{N}\Delta\right], \text{ where } i = 1, 2, ..., N. \tag{6}$$

$\Delta$ is the sampling interval, which is set to $\Delta_{\text{init}} = (t_f - t_n)/2$ at the beginning, a half of the distance between near $t_n$ and far bounds $t_f$. And it will decrease monotonically with an exponential decay rate $\gamma$ until it drops to a pre-defined minimal value $\Delta_{\text{min}}$. Formally, $\Delta_n = \max(\Delta_{\text{init}} \exp(-\gamma n), \Delta_{\text{min}})$ for $n$-th decaying step. Additionally, during training when the estimated $t_s$ is too close to the near or the far bound so that the sampling region $[t_s - \Delta, t_s + \Delta]$ exceeds the original range $[t_n, t_f]$, we will shift the region $[t_s - \Delta, t_s + \Delta]$ back to within $[t_n, t_f]$. As shown in Fig. 2, at the beginning of training, points sampled for Eq.(4) will cover the entire volume, leading to dispersed gradients which facilitate the convergence of GOF. And as the training goes, the predicted surface will become more and more accurate, which is the outcome of gradually refining the sampling region, and in turn also makes the above shrinking operation valid.

Thanks to the dedicated shrinking process, the color weights $w_i$ can successfully concentrate on the object surface as illustrated in Fig. 1(a). As a result, GOF is capable of synthesizing high-fidelity images in a 3D-consistent manner and simultaneously capturing compact object surfaces. During inference, to synthesize an image under a random camera pose $\xi \sim p_\xi$, the generator $g_\theta$ will fetch a truncated latent code $\hat{\mathbf{z}}$ and sample $N$ points $\{\mathbf{x}_i\}$ on each ray within the minimal region $[t_s - \Delta_{\text{min}}, t_s + \Delta_{\text{min}}]$ for the rendering as in Eq.(4). An important benefit of learning a compact object surface is that we can effectively reduce the number of sampled points for rendering, even using only one point on each ray, *i.e.* the surface point. As shown in Fig. 1 (b), the image rendered with only the surface point is almost indistinguishable from that with multiple points. Such equivalence can be guaranteed theoretically when $\Delta_{\text{min}} \to 0$, and we include the proof in the supplementary material.

### 3.3 Loss Functions

Instead of training on posed 2D images, the proposed GOF leverages a corpus of unposed images for 3D-aware image synthesis, where multiple loss functions are adopted.

**GAN Loss.** Following pi-GAN [6], a GAN loss is used where GOF synthesizes fake images by randomly sampling camera poses $\xi$ from a dataset-related distribution $p_\xi$ and rendering according to Eq.(4). Denote $I$ as a real image from the data distribution $p_\mathcal{D}$, the non-saturating GAN loss can be described as follows:

$$
\begin{aligned}
\mathcal{L}_{\text{origin}}(\theta_D, \theta_G) = \mathbf{E}_{\mathbf{z} \sim p_\mathbf{z}, \xi \sim p_\xi} &\left[ f\Big( D_{\theta_D}(G_{\theta_G}(\mathbf{z}, \xi)) \Big) \right] \\
&+ \mathbf{E}_{I \sim p_\mathcal{D}} \left[ f(-D_{\theta_D}(I)) + \lambda |\nabla D_{\theta_D}(I)|^2 \right], \\
\text{where } f(u) &= -\log(1 + \exp(-u)).
\end{aligned}
\tag{7}
$$

However, $\mathcal{L}_{\text{origin}}$ alone is not sufficient to guide the training, which may lead to messy images with smoke-like artifacts. Therefore, two more regularizations are incorporated to reduce artifacts and further smooth the learned surfaces.

**Normal Regularization.** The first regularization is a prior on the surface normal smoothness, which is specially useful for learning from 2D real-world images [11]. In GOF, this normal prior is only employed for the surface points $\mathbf{x}_s \in \mathcal{S}$ to encourage a natural and smooth surface:

$$
\mathcal{L}_{\text{normal}} = \sum_{\mathbf{x}_s \in \mathcal{S}} ||\mathbf{n}_\theta(\mathbf{x}_s, \mathbf{z}) - \mathbf{n}_\theta(\mathbf{x}_s + \boldsymbol{\epsilon}, \mathbf{z})||_2,
\tag{8}
$$

where $\boldsymbol{\epsilon}$ is a small random 3D perturbation and $\mathbf{n}_\theta$ denotes the normal vector, which can be computed by $\mathbf{n}_\theta(\mathbf{x}, \mathbf{z}) = \nabla_\mathbf{x} \alpha_\theta(\mathbf{x}, \mathbf{z}) / ||\nabla_\mathbf{x} \alpha_\theta(\mathbf{x}, \mathbf{z})||_2$.

**Opacity Regularization.** Since alpha values predicted in GOF can be regarded as occupancy values, ideally the entropy of them should be 0 so that $\alpha_\theta(\mathbf{x}, \mathbf{z})$ values will equal 1 for points in the occupied space and 0 for points in the free space. We thus apply the second opacity regularization, aiming to reduce the entropy of predicted alpha values:

$$
\mathcal{L}_{\text{opacity}} = \frac{1}{N} \sum_{i=1}^{N} \log(\alpha_\theta(\mathbf{x}_i, \mathbf{z})) + \log(1 - \alpha_\theta(\mathbf{x}_i, \mathbf{z})).
\tag{9}
$$

In summary, the final loss function for training GOF can be written as:

$$
\mathcal{L}(\theta, \phi) = \mathcal{L}_{\text{origin}}(\theta, \phi) + \lambda_{\text{normal}} \mathcal{L}_{\text{normal}} + \lambda_{\text{opacity}} \mathcal{L}_{\text{opacity}},
\tag{10}
$$

where $\lambda_{\text{normal}}$ and $\lambda_{\text{opacity}}$ are both balancing coefficients.

## 4 Experiments

**Implementation Details.** Unless stated otherwise, in all experiments we set $N$, the number of points sampled for rendering, to 12, and set $M$, the number of bins used in root-finding, to 12. As discussed in Sec.3.2, we apply an iterative process in root-finding. In practice, the number of iterations is set to $m_s = 3$ times. During inference, GOF requires $M + m_s + N$ queries to obtain the color of a pixel, while existing methods require $2N$ queries due to the use of a hierarchical sampling strategy. Recall it is sufficient for GOF to sample only the surface point to render an image, GOF is thus capable of using just $M + m_s + 1$ queries, potentially speeding up the rendering process in off-line applications. More training and implementation details can be found in the supplemental material.

**Datasets.** To assess our method comprehensively, we conduct experiments on three datasets, namely CelebA [44], BFM [45], and Cats [46]. Specifically, CelebA is a high-resolution face dataset containing $200,000$ diverse face images. Following pi-GAN [6], we crop all images in CelebA from the top of the hair to the bottom of the chin as a pre-processing step. As for the Cats dataset, it contains $6,444$ cat faces of size $128 \times 128$. Finally, BFM is a synthetic face dataset rendered with Basel Face Model, where each face is paired with a ground-truth depth map, making it a good benchmark for quantitatively evaluating the quality of learned surfaces.

**Comparison with baselines.** To validate the effectiveness of GOF, we compare it with two representative GRAF methods, namely GRAF [7] and pi-GAN. Firstly, Fig. 3 demonstrates the qualitative

Table 1: **Quantitative results** (128 × 128 px) on BFM, CelebA and Cats datasets, on three metrics Fréchet Inception Distance (FID), Inception Score (IS) and the weighted variance of sampled depth $\Sigma_{t_i}(\times 10^{-4})$.

| | BFM | | | CelebA | | | Cats | | |
|---|---|---|---|---|---|---|---|---|---|
| | FID↓ | IS↑ | $\Sigma_{t_i}$ ↓ | FID↓ | IS↑ | $\Sigma_{t_i}$ ↓ | FID↓ | IS↑ | $\Sigma_{t_i}$ ↓ |
| GRAF [7] | 45.2 | 1.49 | 4.64 | 41.4 | 1.86 | 6.51 | 28.6 | 1.65 | 5.47 |
| pi-GAN [6] | 16.4 | 2.49 | 8.16 | 15.1 | 2.63 | 14.58 | 16.6 | 2.09 | 5.91 |
| Ours (w/o $\mathcal{L}_{normal}$) | 15.6 | 2.85 | 2.66 | 17.0 | 2.29 | 7.15 | 16.1 | 1.92 | 6.06 |
| Ours (w/o $\mathcal{L}_{opacity}$) | 17.1 | 2.61 | 3.37 | 14.4 | **2.91** | 4.94 | 14.3 | 2.35 | 4.42 |
| Ours | **15.3** | **2.89** | **2.54** | **14.2** | 2.87 | **4.58** | **14.1** | **2.48** | **4.28** |

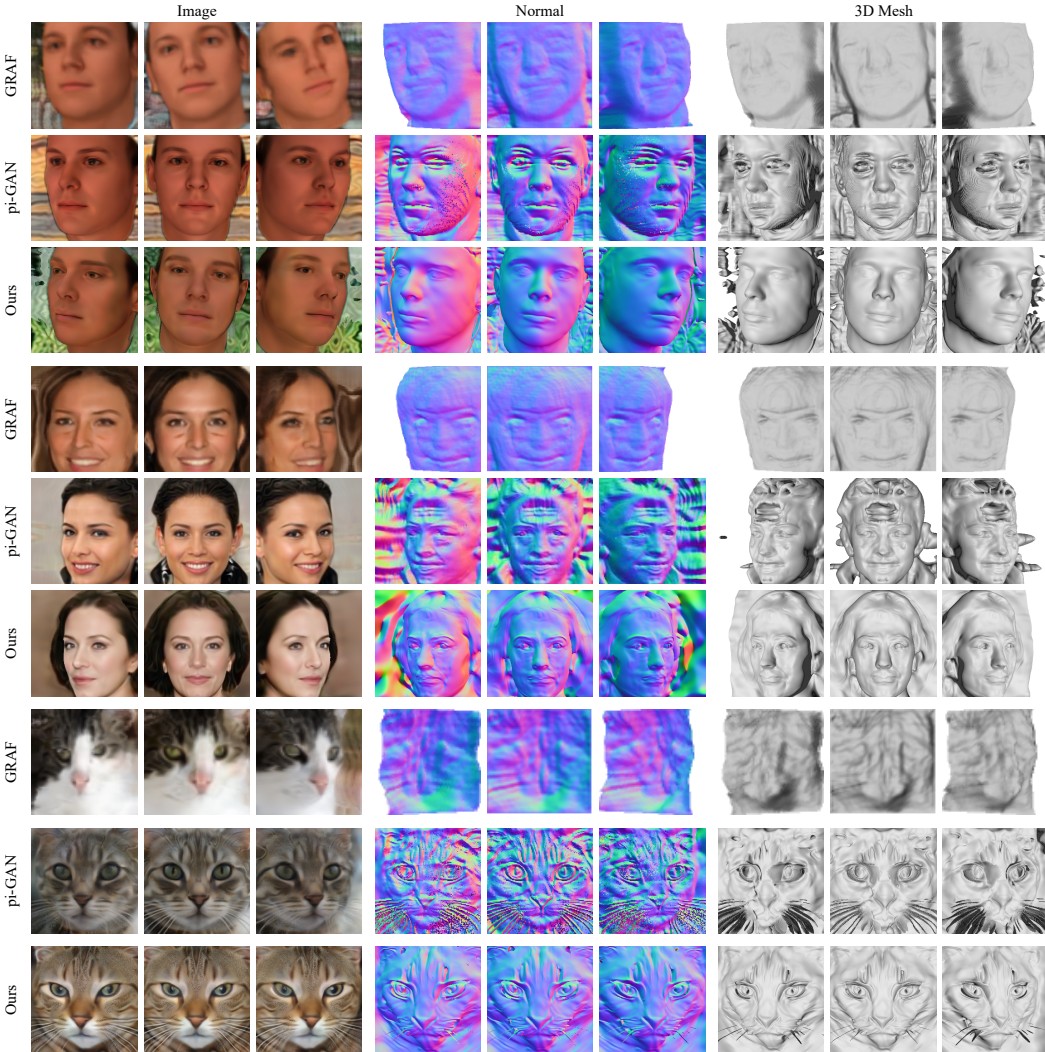

Figure 3: **Qualitative comparison** on BFM (top), CelebA (middle), and Cats (bottom) datasets. Our method synthesizes realistic images while ensuring compact object surfaces.

comparison between these three methods, where we include the synthesized images, the learned surfaces in the form of 3D meshes, as well as the corresponding normal maps. As can be observed, GRAF struggles to render good images, let alone estimate compact and reasonable underlying surfaces. Compared to GRAF, pi-GAN can synthesize images and estimate corresponding surfaces with improved quality. However, messy parts can be clearly recognized on its learned surfaces and normal maps, indicating it is incapable of capturing the compact 3D geometric details. In contrast to both pi-GAN and GRAF, the proposed GOF is shown to hallucinate realistic images with 3D consistency

Table 2: Comparisons on the compactness and accuracy of learned surfaces.

| Method | SIDE ↓ | MAD ↓ |
|---|---|---|
| Supervised | 0.412 | 10.84 |
| Unsup3d [37] | 0.795 | 16.51 |
| GRAF [7] | 1.866 | 26.69 |
| pi-GAN [6] | **0.727** | 20.46 |
| GAN2Shape [38] | 0.759 | 14.94 |
| Ours | 0.779 | **13.81** |

Table 3: Comparisons on the geometry properties of learned surfaces. We report **mean curvature(MC)**$(\times 10^{-3})$ and **mean geodesic distance(MGD)** between random points to assess the geometry properties of recovered surfaces.

| | | BFM | CelebA | Cats |
|---|---|---|---|---|
| MC ↓ | pi-GAN | 16.84 | 25.94 | 34.05 |
| | Ours | **12.25** | **23.13** | **30.14** |
| MGD ↓ | pi-GAN | 0.483 | 0.450 | 0.494 |
| | Ours | **0.226** | **0.231** | **0.317** |

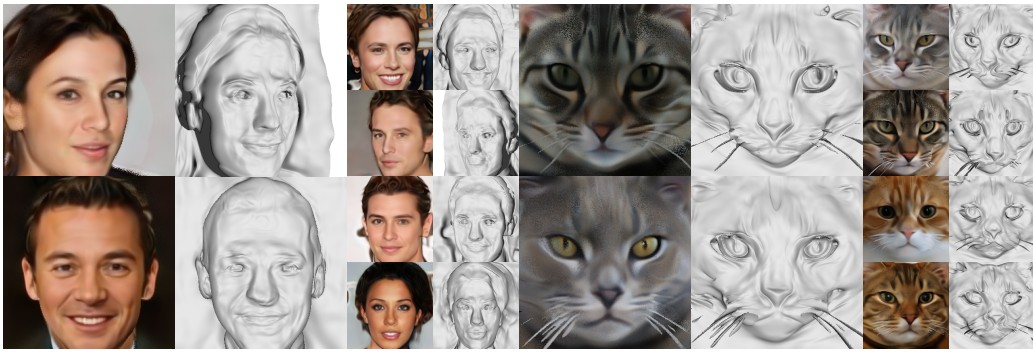

Figure 4: Generated images and their 3D meshes on CelebA and Cats datasets.

and simultaneously learn smooth surface normals as well as compact object surfaces, which verifies the benefit of adopting the transition from radiance fields to occupancy fields. More qualitative results of synthesized images and corresponding surfaces are included in Fig. 4.

To quantitatively evaluate the quality of generated images, we report the Fréchet Inception Distance (FID) scores and Inception Score (IS) scores in Table 1. On these two metrics, GOF demonstrates substantial improvements over baseline methods. To further measure the compactness of learned surfaces, the concentration of color weights $w_i$ as mentioned in Fig. 1 (a) is also computed. Specifically, We sample $N = 36$ equally-spaced points $\{\mathbf{x}_i = \mathbf{o} + t_i\mathbf{d}\}$ within near and far bounds $[t_n, t_f]$ and calculate the corresponding color weights $w_i, i = 1, 2, ..., N$. Actually, the weighted variance of these samples' depth $t_i$ *reflects* the concentration of color weights in a single image:

$$\Sigma_{t_i} = \frac{N}{(N-1)\sum_{i=1}^{N} w_i} \sum_{i=1}^{N} w_i(t_i - \bar{t})^2, \text{ where } \bar{t} = \sum w_i t_i \Big/ \sum w_i.$$

Intuitively, a smaller variance implies the learned surface is more compact. Finally, for each method, the overall concentration of color weights is averaged over 1000 randomly synthesized images at the $256 \times 256$ resolution. The results in terms of this new metric are also included in Table 1.

For the quality of learned surfaces, we first evaluate the compactness and accuracy of surfaces on the BFM dataset, since it contains ground-truth depth maps. Specifically, $50K$ images are generated by each method, together with their corresponding depth maps. For each method, we train a separate CNN on these generated images and depth maps to predict depths from images. Subsequently, we can measure the accuracy of learned surfaces by running the CNN on the test split of BFM and comparing its outputs to the ground-truth depth maps using the scale-invariant depth error (SIDE) and the mean angle deviation (MAD). While MAD focuses more on the compactness of surfaces, SIDE emphasizes more on the accuracy of depth. As shown in Table 2, GOF significantly outperforms baseline methods on the MAD metric and is comparable to strong baselines on the SIDE metric. Moreover, we also report mean curvature (**MC**) and mean geodesic distance (**MGD**) between random points to assess the geometry properties of learned surfaces. The lower these two metrics, the smoother recovered object surfaces. Owing to the absence of such two metrics on ground-truth surfaces for reference, we consider the smoother surfaces better conform to ground-truth cases. The reported values on these two metrics are averaged over 100 randomly synthesized 3D meshes. Quantitative comparisons in Table 3 demonstrate our method GOF can preserve better geometry properties.

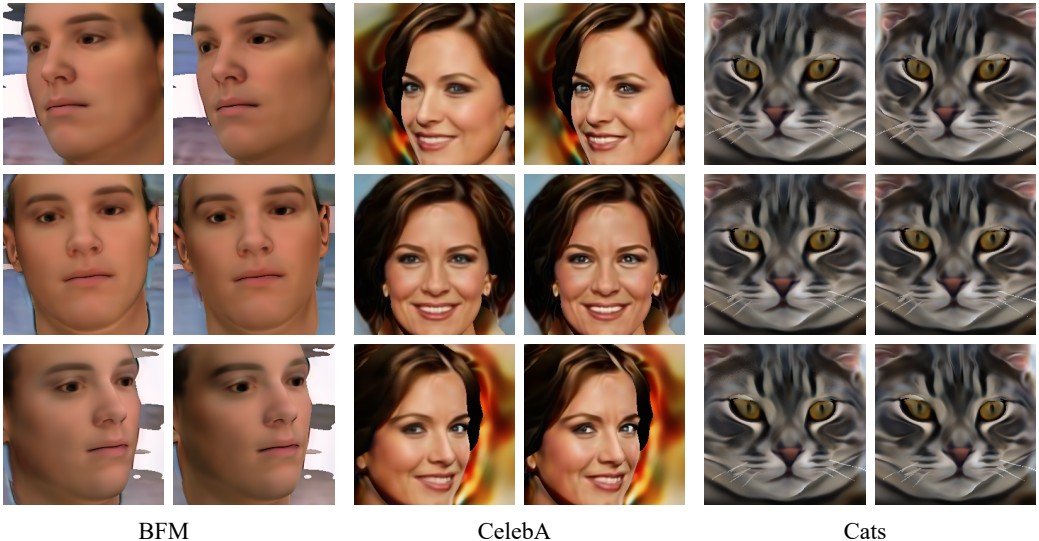

BFM                 CelebA                 Cats

Figure 5: **Rendering with only surface points.** Images (right) rendered only with surface points are indistinguishable from those (left) obtained with cumulative rendering.

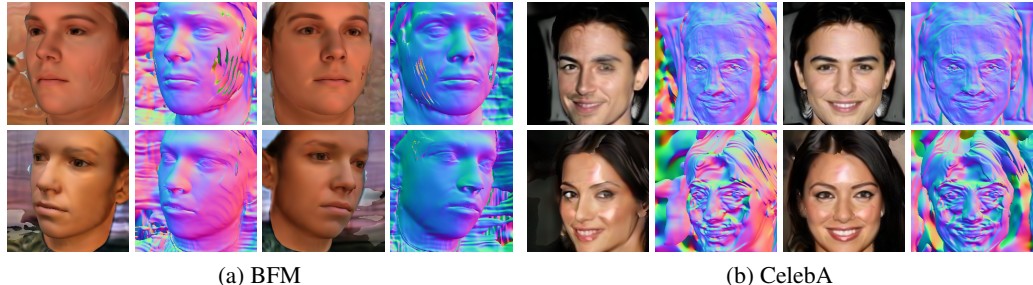

(a) BFM                              (b) CelebA

Figure 6: **Qualitative ablation on proposed priors** (upper row w/o $\mathcal{L}_{\text{opacity}}$, bottom row w/o $\mathcal{L}_{\text{normal}}$).

**Rendering only with surface points.** As mentioned in Sec. 3.2, GOF is able to render an image using only the surface points. To verify this, we showcase in Fig. 5 images rendered by GOF using multiple points and only the surface point. As can be observed, images synthesized with these two strategies are nearly indistinguishable from each other. Thus, GOF possesses the potential to significantly reduce the number of generator queries when synthesizing an image. To compare the efficiency straightforwardly, we estimate the rendering speed of $256 \times 256$ images for both pi-GAN and GOF on a single Intel Xeon(R) CPU. On average, pi-GAN costs about 78s per image, while GOF takes about 56s, saving approximately $28\%$ of the time. Owing to the reduction in the burden of queries, GOF enables a light rendering scheme that is promising for applications on mobile devices.

**Ablation studies.** We here analyze the effects of the proposed regularizations $\mathcal{L}_{\text{normal}}$ and $\mathcal{L}_{\text{opacity}}$. Table 1 includes the quantitative ablation study on these priors. We also include qualitative samples in Fig. 6, which contains images synthesized by GOF without one regularization item. As shown in the BFM cases 6(a), removing opacity prior leads to the smoke-like artifacts around the cheek part and the absence of normal regularization might degrade the quality of learned normal maps. While testing on the real-world dataset 6(b), undesirable specular highlights emerge on the face and the hollows appear on the corresponding shapes if without the normal regularization. Moreover, we observe that removing opacity prior on CelebA dataset will make the face surfaces too flat and unnatural. It is worth noting that although the performance of GOF is deteriorated due to the absence of these priors, images and surfaces produced by GOF are still of reasonable quality when compared to that from previous approaches, indicating the transition from radiance fields to occupancy fields is the main cause that leads to the success of GOF. Moreover, we showcase the degenerated results on BFM dataset if our model is trained without the shrinking process. As illustrated in Fig. 7, despite the realistic generated images, there emerges random noise on the corresponding normal maps and some nasty dents appear on the face shapes, which demonstrates that the combination of our proposed occupancy representation and the shrinking sampling procedure ensures the surface compactness.

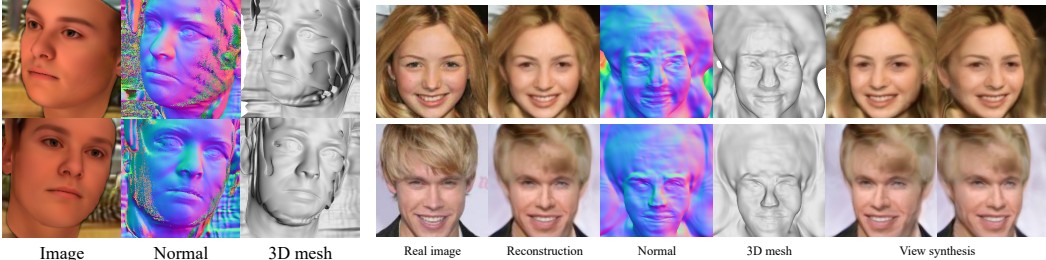

| Image | Normal | 3D mesh | Real image | Reconstruction | Normal | 3D mesh | View synthesis |

Figure 7: **GOF results without the shrinking process.** Noise emerges on normals and dents appear on shapes.

Figure 8: **GAN inversion results on real images.** GOF can reconstruct the target images and simultaneously learn the corresponding normal maps as well as 3D shapes.

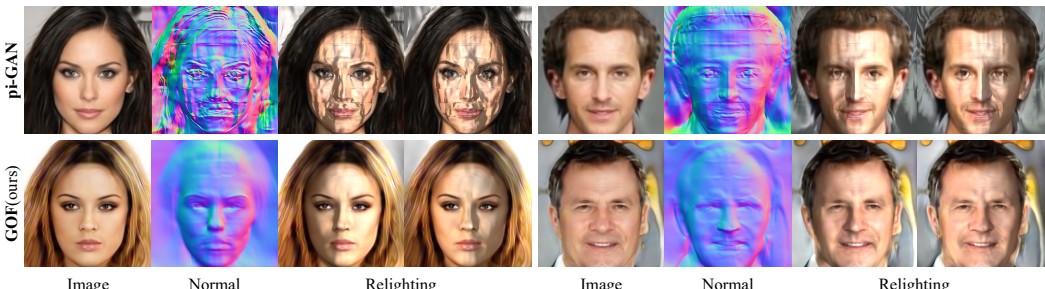

| Image | Normal | Relighting | Image | Normal | Relighting |

Figure 9: **Relighting results.** Our method GOF generates desirable images under various light conditions while baseline results are far from satisfactory.

**Inverse rendering.** Through GAN inversion, our method is also capable of inverse rendering as shown in Fig. 8. Given a real image, GOF can reconstruct the target image successfully and realize free view synthesis by controlling the viewpoints. Besides, the recovered normal maps as well as 3D shapes pave the way for downstream tasks such as relighting and editing.

**Relighting.** In Fig. 9 we provide the relighting results based on the learned normal maps by explicitly controlling the lighting directions. As our method and baselines can't predict the corresponding albedo, the face-forwarding image is considered as the pseudo albedo. Thanks to better learned normal maps, our method GOF presents promising images under different light conditions. In contrast to ours, baseline methods like pi-GAN [6] tend to generate messy normal maps with obvious checkerboard-like artifacts, leading to noisy and dissatisfied relighting results.

**Limitations.** While training on real-world datasets, our method GOF might present similar dents in the hair regions as in existing approaches [6]. Besides, the adopted FiLMed-SIREN backbone in the generator will lead to stripe artifacts in the generated images especially when they are rendered only with surface points. Meanwhile, surface rendering mode will make furry cat images over-smooth and less realistic. Moreover, our method is more suitable for solid objects with only one surface.

## 5    Conclusion

In this work, we propose generative occupancy fields (GOF), a novel generative radiance fields for 3D-aware image synthesis. The crux of GOF is a dedicated transition from the cumulative rendering in radiance fields to rendering with only the surface points. Such a transition is inspired by the resemblance between the alpha values in radiance fields and the occupancy values in occupancy networks, so that we can reinterpret one as the other. In practice, such a transition is achieved during training by gradually shrinking the sampling region in the rendering process of GOF from the entire volume to a minimal neighboring region around the surface, where the surface is located via root-finding on predicted alpha values. Thanks to the transition, surfaces learned by GOF continuously converge during the training, ensuring their compactness at the end. On three diverse datasets, GOF is shown to demonstrate great superiority in synthesizing 3D consistent images and in the meantime capturing compact surfaces, significantly broadening the application of generative radiance fields in downstream tasks.

## Acknowlegements

We would like to thank Eric R. Chan for sharing the codebase of pi-GAN. This work is supported by the Collaborative Research Grant from SenseTime (CUHK Agreement No. TS1712093), the General Research Fund (GRF) of Hong Kong (No. 14205719), the RIE2020 Industry Alignment Fund – Industry Collaboration Projects (IAF-ICP) Funding Initiative, as well as cash and in-kind contribution from the industry partner(s).

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
