# Generative Occupancy Fields for 3D Surface-Aware Image Synthesis
# (Supplementary Material)

Here we provide implementation details, additional results on CARLA dataset, and proof of the equivalence between two rendering schemes when $\Delta_{\min} \to 0$. A brief discussion on the future works and broader impacts is also included. Our code and models are available at `https://github.com/SheldonTsui/GOF_NeurIPS2021`.

## A    Model Details

Following StyleGAN [1], the mapping network is an MLP with three hidden layers of 256 units each. Besides, we leverage the FiLMed-`SIREN` [2] module as the backbone for the generator $G_\theta$ [3]. On the head of predicting $\alpha(\mathbf{x}, \mathbf{z})$, a sigmoid function is included to ensure the value range.

Similar to pi-GAN [3], our discriminator $D_\theta$ grows progressively as training goes. The resolution of training images is initially set as $32 \times 32$ and doubled twice during training, up to $128 \times 128$. Apart from discriminating the generated images, the discriminator $D_\theta$ will additionally predict the corresponding latent code $\hat{\mathbf{z}}$ and the camera pose $\hat{\xi}$, which will be used to compare with the ground-truth values as additional losses.

## B    Additional Training Details

For all datasets used in the experiments, we assume a pinhole perspective camera with a field of view of $12°$. During training, we sample camera poses $\xi$ from a Gaussian distribution $p_\xi$ for BFM and CelebA dataset. For Cats dataset, a uniform distribution is leveraged as the setting in pi-GAN [3]. During training, the opacity coefficient $\lambda_{\text{opacity}}$ will grow monotonically with an exponential rate $\gamma_{\text{opac}}$ following $\lambda_{\text{opacity}} = \min(\lambda_{\text{opac\_init}} \cdot \exp(n \gamma_{\text{opac}}), 10)$. When computing the surface normals, we set the Euclidean norm of the small random 3D perturbation $\epsilon$ as $0.01$. Besides, we find the hierarchical sampling is still effective in our method. For a fair comparison with baseline methods, we uniformly set the number of bins in root-finding $M$ to 9, set the number of coarse samples $N_{\text{coarse}}$ to 9 and set the number of fine samples $N_{\text{fine}}$ to 6 in our method. In Table 1 we include the values of important dataset-dependent hyperparameters of GOF.

Table 1: The setting of several important dataset-dependent hyperparameters.

| dataset | $\gamma$ | $t_n$ | $t_f$ | $\Delta_{\min}$ | $\lambda_{\text{normal}}$ | $\lambda_{\text{opac\_init}}$ | $\gamma_{\text{opac}}$ | $\sigma_v$ | $\sigma_h$ |
|---|---|---|---|---|---|---|---|---|---|
| BFM | $4.0 \times 10^{-5}$ | 0.88 | 1.12 | 0.01 | 0.002 | 0.1 | $4.0 \times 10^{-5}$ | 0.155 | 0.3 |
| CelebA | $1.0 \times 10^{-5}$ | 0.88 | 1.12 | 0.03 | 0.05 | 0.01 | $0.5 \times 10^{-5}$ | 0.155 | 0.3 |
| Cats | $2.0 \times 10^{-5}$ | 0.8 | 1.2 | 0.1 | 0.05 | 0.02 | $1.0 \times 10^{-5}$ | 0.4 | 0.5 |

Our models are trained on 8 TITAN XP GPUs on all datasets. The whole training process on BFM, CelebA and Cats takes about 26 hours, 66 hours and 12 hours respectively. To avoid the hollow face illusion [4], the training of all models starts from an early (about 2K iterations) pretrain model with the correct outward-facing faces. Owing to the change of image resolution during training, the

corresponding batch size and learning rate will be adjusted accordingly. In Table 2 we list the values of these hyperparameters across different datasets.

Table 2: The setting of several hyperparameters to be adjusted during training.

| Training Stage (iterations) | | | batch size | resolution | $lr(G_\theta)$ | $lr(D_\theta)$ |
|---|---|---|---|---|---|---|
| BFM | CelebA | Cats | | | | |
| $0 \sim 10K$ | $0 \sim 20K$ | $0 \sim 5K$ | 128 | 32 | $5.0 \times 10^{-5}$ | $2.0 \times 10^{-4}$ |
| $10K \sim 60K$ | $20K \sim 160K$ | $5K \sim 30K$ | 64 | 64 | $5.0 \times 10^{-5}$ | $2.0 \times 10^{-4}$ |
| $60K \sim 80K$ | $160K \sim 200K$ | $30K \sim 40K$ | 32 | 128 | $4.0 \times 10^{-6}$ | $2.0 \times 10^{-5}$ |

Due to the absence of root-finding [5], baseline methods such as GRAF [6] and pi-GAN [3] have to regard the weighted depth in the cumulative rendering process as the final predicted depth. For a specific ray $\mathbf{r} = \mathbf{o} + t\mathbf{d}$ with $N$ sampled points $\{\mathbf{x}_i = \mathbf{o} + t_i\mathbf{d}\}$, the depth $\bar{t}_s$ is estimated as follows:

$$\bar{t}_s = \sum_{i=1}^{N} w_i t_i = \sum_{i=1}^{N} \exp\big(-\sum_{j<i} \sigma_\theta(\mathbf{x}_j, \mathbf{z})\delta_j\big)\big(1 - \exp(-\sigma_\theta(\mathbf{x}_i, \mathbf{z})\delta_i)\big)t_i. \tag{1}$$

## C   Equivalence Proof

As mentioned in Sec. 3.2, we include two different rendering schemes during inference. We here demonstrate the equivalence of these two schemes when $\Delta_{\min} \to 0$. For each ray $\mathbf{r} = \mathbf{o} + t\mathbf{d}$, the surface point $\mathbf{x}_s = \mathbf{o} + t_s\mathbf{d}$ will be firstly determined via root-finding. For the rendering with Eq. 4 of the main paper, we will sample $N$ points $\{\mathbf{x}_i = \mathbf{o} + t_i\mathbf{d}; i = 1, 2, ..., N\}$ within the minimal region around the surface $[t_s - \Delta_{\min}, t_s + \Delta_{\min}]$. Therefore, the cumulative color on the ray $\mathbf{r}$ can be represented as follows:

$$\hat{\mathbf{C}}_c(\mathbf{r}) = \sum_{i=1}^{N} \alpha_\theta(\mathbf{x}_i) \prod_{j<i} \big(1 - \alpha_\theta(\mathbf{x}_j)\big)\mathbf{c}_\theta(\mathbf{x}_i, \mathbf{d}), \tag{2}$$

where the latent code $\mathbf{z}$ is omitted for brevity.
In the implementation, we force the sum of color weights $w_i = \alpha_\theta(\mathbf{x}_i) \prod_{j<i} \big(1 - \alpha_\theta(\mathbf{x}_j)\big)$ to be 1 by letting $w_N = 1 - \sum_{j=1}^{N-1} w_j$. Hence, Eq. 2 can be reformulated to:

$$\hat{\mathbf{C}}_c(\mathbf{r}) = \sum_{i=1}^{N-1} \alpha_\theta(\mathbf{x}_i) \prod_{j<i} \big(1 - \alpha_\theta(\mathbf{x}_j)\big)\big(\mathbf{c}_\theta(\mathbf{x}_i, \mathbf{d}) - \mathbf{c}_\theta(\mathbf{x}_N, \mathbf{d})\big) + \mathbf{c}_\theta(\mathbf{x}_N, \mathbf{d}). \tag{3}$$

For the rendering only with surface points, we have rendered color as $\hat{\mathbf{C}}_s(\mathbf{r}) = \mathbf{c}_\theta(\mathbf{x}_s, \mathbf{d})$. Without loss of generality, we just consider the case of single color channel, *i.e.*, $c_\theta(\mathbf{x}, \mathbf{d}) \in \mathbb{R}$.

**Theorem 1.** *Assuming the color is predicted by the Multilayer Perceptrons with SIREN or ReLU activation functions, we have*

$$\lim_{\Delta_{min} \to 0} \hat{\mathbf{C}}_c(\mathbf{r}) = \hat{\mathbf{C}}_s(\mathbf{r}). \tag{4}$$

*Proof.* Note that, Linear layers, SIREN and ReLU activation functions as well as encoding function in the positional encoding are all Lipschitz continuous thus:

$$|c_\theta(\mathbf{x}_i, \mathbf{d}) - c_\theta(\mathbf{x}_s, \mathbf{d})| \le k'_c ||\mathbf{x}_i - \mathbf{x}_s||_2 \le k_c \Delta_{\min}. \tag{5}$$

Moreover, we further omit $\mathbf{d}$ in $c_\theta(\mathbf{x}, \mathbf{d})$ and have:

$$\left| \hat{\mathbf{C}}_c(\mathbf{r}) - \hat{\mathbf{C}}_s(\mathbf{r}) \right| = \left| \sum_{i=1}^{N-1} \alpha_\theta(\mathbf{x}_i) \prod_{j<i} \left(1 - \alpha_\theta(\mathbf{x}_j)\right) \left(c_\theta(\mathbf{x}_i) - c_\theta(\mathbf{x}_N)\right) + c_\theta(\mathbf{x}_N) - c_\theta(\mathbf{x}_s) \right| \quad (6)$$

$$\leq \sum_{i=1}^{N-1} \alpha_\theta(\mathbf{x}_i) \prod_{j<i} \left(1 - \alpha_\theta(\mathbf{x}_j)\right) |c_\theta(\mathbf{x}_i) - c_\theta(\mathbf{x}_N)| + |c_\theta(\mathbf{x}_N) - c_\theta(\mathbf{x}_s)| \quad (7)$$

$$\leq (N-1)|c_\theta(\mathbf{x}_i) - c_\theta(\mathbf{x}_N)| + |c_\theta(\mathbf{x}_N) - c_\theta(\mathbf{x}_s)| \quad (8)$$

$$< 2k_c N \Delta_{\min}. \quad (9)$$

where inequality 8 holds by $0 \leq \alpha_\theta(\mathbf{x}) \leq 1$. Therefore, for any $\epsilon > 0$, we set $\Delta_{\min} = \epsilon/2k_c N$ and have:

$$\left| \hat{\mathbf{C}}_c(\mathbf{r}) - \hat{\mathbf{C}}_s(\mathbf{r}) \right| < 2k_c N \Delta_{\min} = \epsilon.$$

$\blacksquare$

# D  Additional Results on CARLA

As presented in GRAF [6] and pi-GAN [3], baselines have already demonstrated remarkable results for both synthesized images and corresponding shapes on CARLA dataset. We also implement our approach GOF on this synthetic dataset and achieve comparable performance in terms of the image quality as provided in Table 3. Despite the satisfying images, baseline methods sometimes generate nasty car shapes with dents on the bonnet. Fig 1 shows such shape artifacts in the normal and depth maps. By contrast, our method can not only synthesize realistic images but also learn good shapes. In the experiments, the aforementioned shrinking process will lead to undesirable occupancy outside the cars and thus be removed here.

Table 3: **Quantitative results** on CARLA dataset, on five different metrics, FID($128 \times 128$ px), IS, $\Sigma_{t_i}(\times 10^{-4})$, MC and MGD.

|  | FID↓ | IS↑ | $\Sigma_{t_i}$ ↓ | MC↓ | MGD↓ |
|---|---|---|---|---|---|
| GRAF [6] | 37.2 | 3.89 | 0.93 | 13.11 | 0.866 |
| pi-GAN [3] | 29.6 | **4.35** | 1.74 | 13.07 | 0.874 |
| Ours | **29.3** | 4.29 | **0.61** | **12.49** | **0.831** |

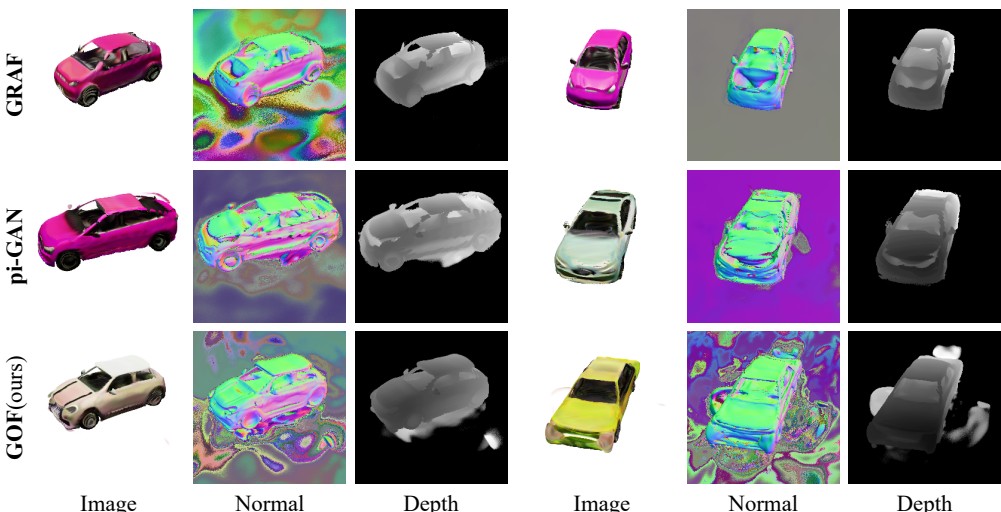

Figure 1: Qualitative comparison on CARLA dataset. Baseline methods predict dents on the car bonnets while ours avoids this issue successfully.

# E Future Works

In the experiments, we discover the trade-off between the FID score and shapes. Following the official code[1] of pi-GAN, we can increase the learning rate and gradient clip range, decrease the R1 regularization on the discriminator, and replace the progressive discriminator to achieve a lower FID score. However, the corresponding shapes will degenerate under this circumstance. We identify exploring how to get rid of such a trade-off as promising future work. Moreover, baseline methods including ours struggle to recover the eyes geometry especially on CelebA dataset. Firstly, the light field of the eyes is more complicated than in other regions. More importantly, the dataset is biased, where people always gaze at the camera when taking photos, the biased eye poses are inadequate to provide multi-view information for modeling eyes accurately. It's also an interesting problem to be mitigated in the future.

# F Broader Impacts

Our work aims at generating images in a 3D consistent manner and simultaneously learn compact and smooth object surfaces. Its application lies mainly in entertainment industries such as AR/VR or video games. However, our framework may be potentially used in the face forgery like DeepFake. Also, computational cost as well as energy consumption should also be considered during the development of such systems for environmental protection.

# G Additional Qualitative Results

In Fig. 2 we include more qualitative results generated by the proposed GOF. Fig. 3 shows that GOF can render high-quality images using only the surface points. In Fig. 4 we present the linearly interpolating results between two latent codes on CelebA [7] and Cats [8] respectively. Moreover, we provide a demo video to demonstrate that the proposed GOF is capable of generating realistic images in a 3D-consistent manner and simultaneously capturing compact object surfaces.

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

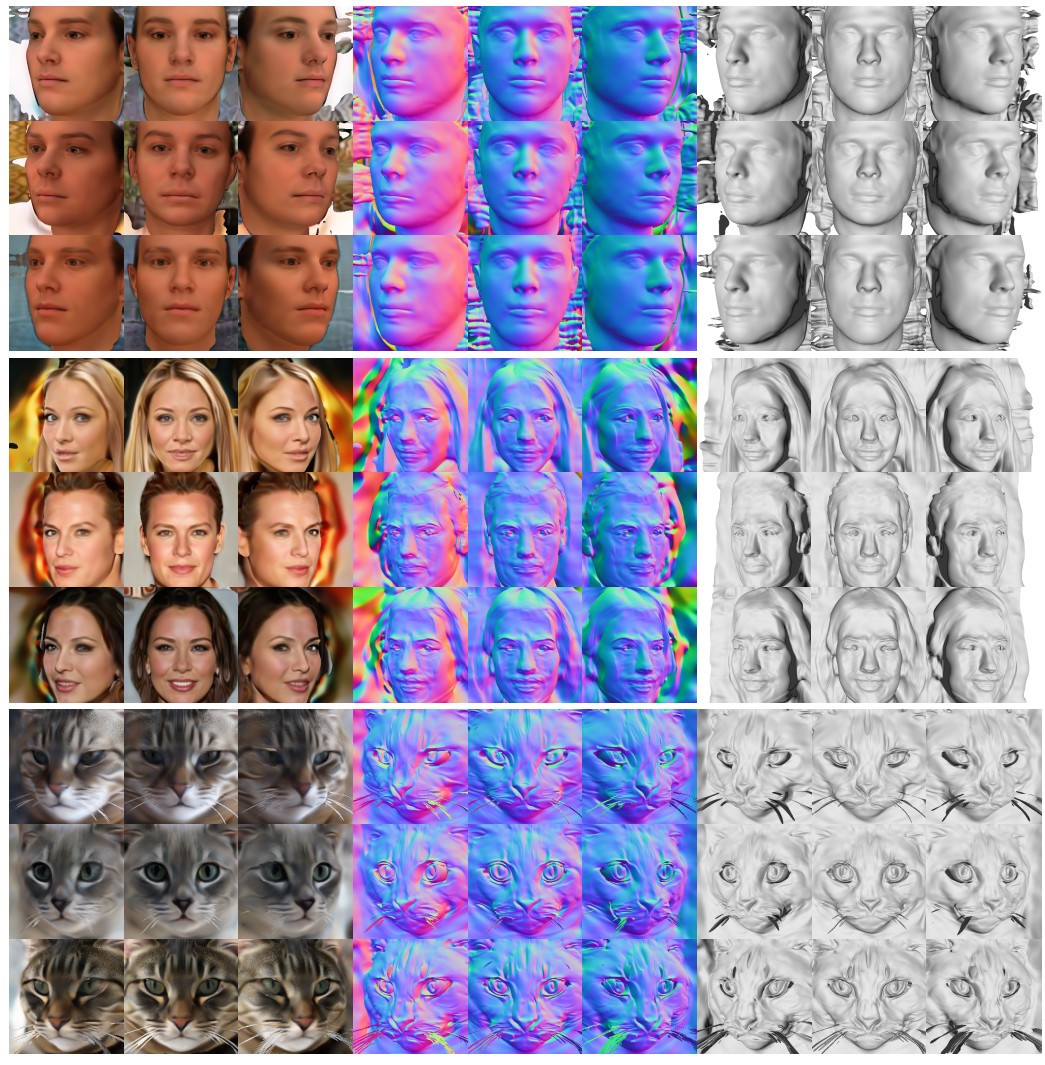

Figure 2: More qualitative results from our model GOF trained on BFM (top), CelebA (middle), and Cats (bottom) datasets.

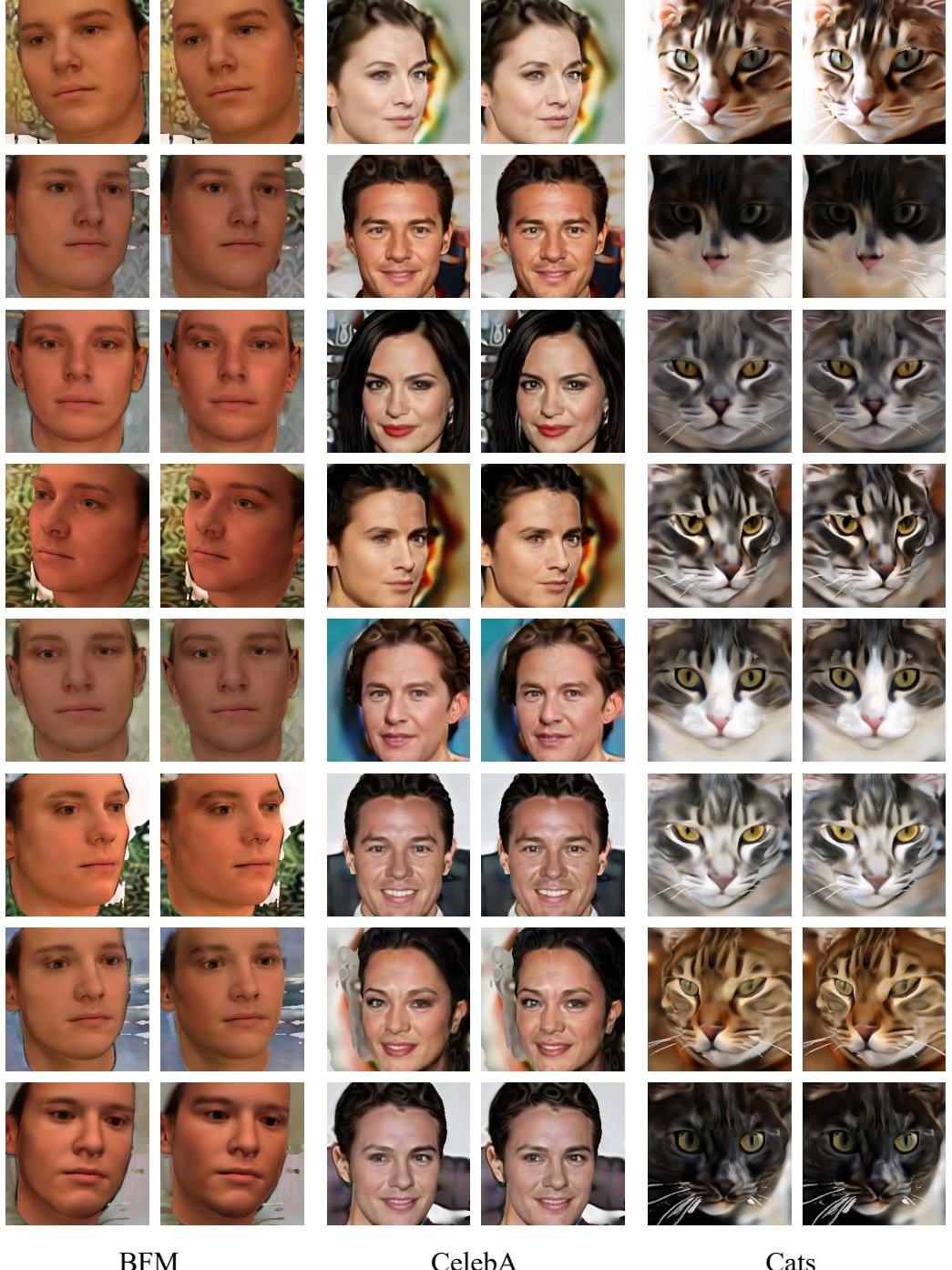

BFM                    CelebA                    Cats

Figure 3: **Rendering only with surface points.** We provide more images rendered only with surface points (right), which are almost indistinguishable from those obtained with cumulative rendering (left).

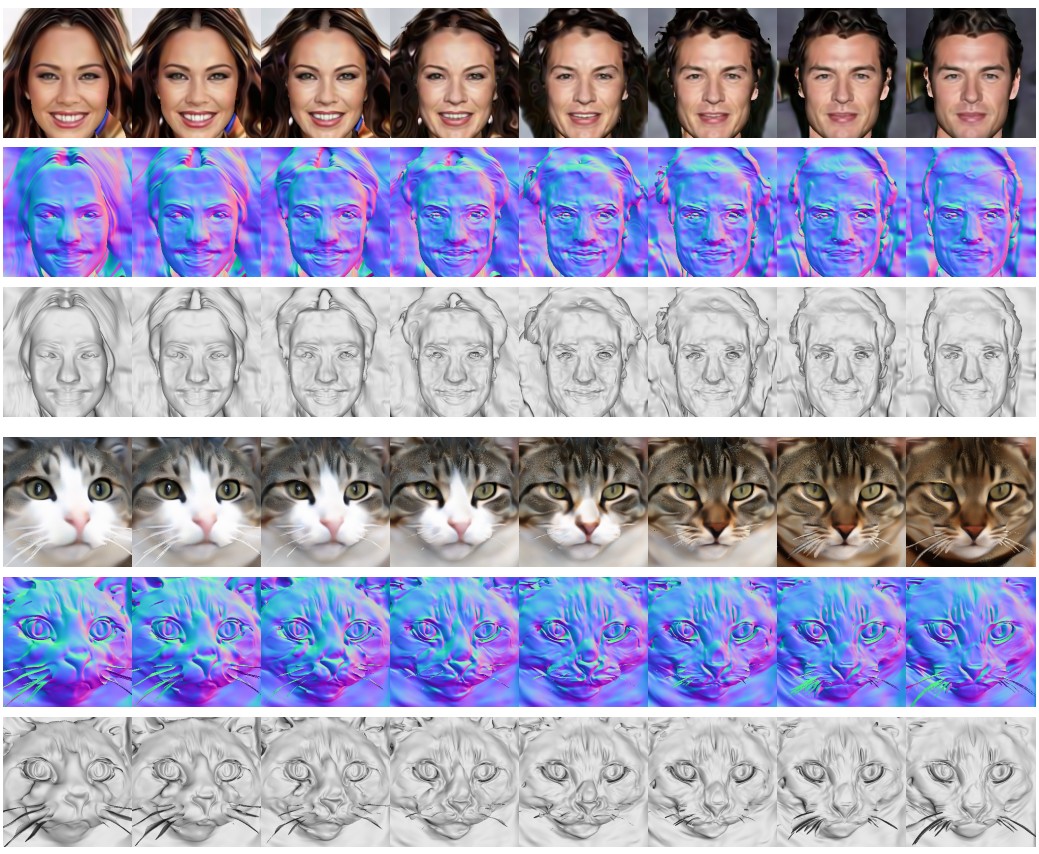

Figure 4: Linearly interpolating between two latent codes on CelebA and Cats datasets.