# OpenReview forum: "Generative Occupancy Fields for 3D Surface-Aware Image Synthesis"
_NeurIPS.cc/2021/Conference — NeurIPS 2021 Poster_

### Official Review · Reviewer_P3Ee · 2021-07-11

**Rating:** 7
**Confidence:** 5

**Summary:**

The authors propose a novel formulation of the 3D GAN problem using hybrid volume-surface approach. The method leads to a more accurate shape reconstruction allowing for more efficient rendering. The reconstructed image quality is also comparable or better.

**Ethical Concerns:**

The authors are concerned with deep fakes and environmental impact. That seems like a fitting assessment.



**Limitations And Societal Impact:**

[Societal impact]

The method could be utilized in fast low-cost content production for 3D applications including AR/VR.


[Limitations]

Some of the results trade off blur and noise for sharp edge artifacts that are subjectively more disturbing (see the eye region in the last cat example in the video). Same in Fig 5.

The authors discuss the limitations in the supplemental which does not seem right. The discussion should clearly be in the main paper as it is essential to the message of the paper. The authors admit shape distortions in some difficult areas. The authors also mention "stripe" artifacts in surface based renderings. I am not sure if this is the same artifact that I observed above but I note that the sharp-edge artifacts are equally present in both the surface and volume renders (see Fig. 5).

The authors could also discuss the resolution as a potential limitation and note that the quality of the generated images is nowhere near to what 2D GANs such as StyleGAN2 can produce (which is true for all exisiting 3D GANs).



**Main Review:**

[Originality]

The method closely follows standard 3D GAN pipeline but introduces novel volume-surface formulation of the representation. The authors explain motivation for the technique and demonstrate its benefits.

[Quality]

The technique is demonstrated adequately and the evaluation is sufficient.

[Clarity]

The exposition is clear. All necessary details for reimplementation appear to be present. The network architecture in the supplemental could be described more visually for intuitive comparison to other methods.

[Significance]

The paper falls within the scope of image/shape GAN methods and shares their potential application scope.

[Questions, Comments and Concerns]

1) The Eq. 4 replaces exponential of sum (of sigmas) by effectively a product of exponentials (as the alphas are computed by sigmoids). These two formulations are mathematically very close. Would the original NeRF formulation (Eq. 2) still be applicable if the exp(-sigma) would be used for mapping into the desired [0, 1] range before thresholding? In another words, is the key innovation the adaptive sampling around the surface alone or is the alpha formulation also necessary?

2) The surface sampling pattern of GOF appears more efficient than that of GRAF. However, the training was still limited to 128x128 px resolution as in pi-GAN [6]. Why is that? Should this be listed in limitations (same as for previous work)?

3) In Sup Fig. 3, the authors demonstrate interpolation in the latent space. However, is the method suitable also for inverse rendering, that is finding a fitting latent code for given output 2D image? Does the surface manifold representation provide robust gradient for such tasks?

4) Minor comments:
- L110: "ensure the compactness of learned object surfaces." - What does this refer to? Size of the trained model? Area of the extracted surface manifold?
- Eq.2 for T_i misses z in the argument
- Figure 3 is very low-res to the point that it is hard to see differences.
- Figure 6 - please label which prior is ablated in which panel.
- The computational resources were promised in the appendix but I did not find them in the supplement.

[Conclusion]

Overall, the methods main contribution is its formulation of the volume/surface representation. The application to GAN problem does not seem to require any particularly unique features when compared to the multiview supervision as demonstrated by the concurrent work [10]. Still, it is a completely novel technique and demonstrating its efficiency in GAN setup is perfectly valid since it generally tends to be harder to train. The method produces superior surface surfaces when compared to previous volume-only based GAN techniques. The image quality is also quantitatively higher although qualitatively the method does trade-off the sampling noise artifacts to sharp boundary artifacts in some images so the visual preference may be highly subjective and vary on case-by-case basis. Weighing all these factors, I lean positive but the authors should correct their limitation discussion by moving it to the main paper and potentially adding new points.


**Time Spent Reviewing:**

6

---

> ### Author Response · Authors · 2021-08-10
> **Feedback to reviewer P3Ee**
>
> **Q1: Which one is the key innovation? The adaptive sampling procedure or the alpha formulation?**
>
> **A1**: Despite the close correlation between original sigma and proposed alpha on the formulation, the occupancy representation inherently ensures the compact object surface. Actually, the combination of both ensures surface compactness.
>
> **Q2: Why the training is still limited to 128x128 px resolution?**
>
> **A2**: There might be some misunderstanding here. GOF can leverage a more efficient surface sampling pattern **during inference**. When rendering images using a single CPU, GOF saves approximately 28% of the time compared to baselines. For training, GOF is not limited to 128x128 resolution, and we intend to train GOF on a higher resolution to further demonstrate its effectiveness. We conduct experiments on 128x128 to ensure a fair comparison, following previous practices. In terms of memory usage, when training with 128x128 images, GOF only costs 3.8G GPU memory, compared to 7G of baseline methods, leading to a reduction of 46%.
>
> **Q3: How about the inverse rendering?**
>
> **A3**: Yes, our method GOF is also suitable for inverse rendering. To show this, we modify the discriminator to be an encoder E by replacing the final classification convolution layer with an FC layer. Meanwhile, we randomly sample the latent code z and the camera pose \xi, generating 20K training images via the generator g. The encoder E is trained on these images for 10K iterations. Given an input 2D image, the trained encoder E predicts the corresponding latent code, with which GOF can recover a compact 3D object shape. The results of inverse rendering will be complemented in the revision.
>
> **Q4: What does “the compactness of learned surfaces” refer to?**
>
> **A4**: It means the compactness of the extracted surface manifold.
>
> Besides, we will fix the typos in Eq.2, replace Fig3 with a high-resolution figure and clarify the annotations in Fig6. The computational resources will be added in the appendix as well.

---

> > ### Comment · Reviewer_P3Ee · 2021-08-13
> > **Feedback**
> >
> > Thank you for your response. Your answers address my concerns and I retain my acceptance recommendation.
> >
> > I have one follow-up question in response to your communication with eXp1 regarding the FID scores. Why were the FID scores not taken directly from previous publications if the conditions appear to be the same e.g. for CelebA (same dataset, resolution)? The difference seem to be relatively substantial in some cases. Can this be somehow addressed in the final revision?

---

> > > ### Author Response · Authors · 2021-08-13
> > > **Thanks for your feedback!**
> > >
> > > Thanks for your suggestions. Owing to the inaccessibility of pi-GAN official code before NeurIPS submission, we report the FID scores from our reproduced code for a fair comparison. We will try to match the reported results with the official code under a fair and consistent setting.

---

### Official Review · Reviewer_eXp1 · 2021-07-12

**Rating:** 7
**Confidence:** 5

**Summary:**

This paper proposes a new framework for generative radiance fields, along the lines of GRAF and pi-GAN, but the authors focus on generating cleaner surfaces using an approach that is closely related to UNISURF. This method makes sense, and the generated shapes look a lot cleaner and less noisy than those generated by pi-GAN and more expressive than GRAF.

**Ethical Concerns:**

An explicit statement on ethics considerations of this work and unconditional 3D image generation in general would be helpful.

**Limitations And Societal Impact:**

The authors discuss and evaluate their limitations.

**Main Review:**

Paper strengths:
+ generative 3D-aware methods are very interesting and challenging, the proposed framework introduces a surface-centric approach which is new
+ the qualitative results, especially the surfaces (i.e., the focus of this work), look really good - a significant improvement over previous work

Paper weaknesses:

- some missing related work:
-> Wang et al., "NeuS: Learning Neural Implicit Surfaces by Volume Rendering for Multi-view Reconstruction" as a concurrent method to UNISURF, which can be optionally cited
-> Yariv et al., "Volume Rendering of Neural Implicit Surfaces" as a concurrent method to UNISURF, which can be optionally cited
-> Sitzmann et al., "Scene Representation Networks", NeurIPS 2019 as one of the very first 2D supervised neural implicit representations
-> Kellnhofer et al., "Neural Lumigraph Rendering", CVPR 2021 as one of the more recent and SOTA surface-based neural rendering frameworks

- method: the proposed method makes sense, but seems like a relatively simply combination of GRAF and UNISURF; the manuscript would very much benefit from a specific and detailed discussion on what's similar and different between the two

- quantitative evaluation: in table 1, GOF outperforms pi-GAN only marginally; that may be okay, because the focus here is on the surface; however, I was surprised when I realized that the FID scores reported here for pi-GAN on CelebA and Cats are 15.8 and 17.6, respectively, whereas they are listed as 14.7 and 16.8 in that paper, which are both lower than the proposed method

- qualitative results: the cats generated by FOG seem considerably smoother and less realistic than those of pi-GAN

Overall, my assessment is that this submission takes some of the important ideas from UNISURF and applies them to a different problem setting, i.e. that of unconditional image generation. It seems very interesting that the very naive UNISURF-like progressive shrinking technique works in a GAN framework; I had my doubts that such an approach would remain stable once the updates became confined to the local region. Therefore, despite this submission being strongly influenced by the core ideas of UNISURF, I think it merits publication.

**Time Spent Reviewing:**

2

---

> ### Author Response · Authors · 2021-08-10
> **Feedback to reviewer eXp1**
>
> **Q1: Some missing related works.**
>
> **A1**: Thanks for pointing them out. We’ll add these related works in the revision. However, it’s worth noting that “NeuS” and “Volume Rendering of NIS” are released **after** the NeurIPS submission.
>
> **Q2: A specific discussion on the similarity and difference between GOF and the combination of GRAF and UNISURF.**
>
> **A2**: We consider UNISURF as an excellent concurrent work since we have started this research project before it has been published.
> Our method GOF is based on GRAF models and targets to mitigate the problem of diffused object surfaces presented in GRAFs. As presented in the related work section (L103-111), GOF shares similar spirits with UNISURF but is different in tasks and focuses. UNISURF experiments on multi-view 3D reconstruction and aims at getting rid of precise object masks during training. By contrast, GOF focuses on a relatively more challenging task of 3D-aware image synthesis. Meanwhile, the integration of radiance fields and occupancy representations is proposed to ensure compact object surfaces.
>
> **Q3: Doubt on the FID score.**
>
> **A3**: It’s worth noting the official code of pi-GAN is inaccessible before the NeurIPS submission. Therefore, all the experiment results are based on our reproduced code. And we fix the settings across all methods to ensure a fair comparison. We will clarify this in the revision.
>
> **Q4: Rendered cats seem considerably smoother and less realistic.**
>
> **A4**: Thanks for pointing it out. Actually, we can adjust the hyperparameter, minimal sampling interval \Delta_min, to generate more furry and realistic cat images.

---

> > ### Comment · Reviewer_eXp1 · 2021-09-10
> > **rebuttal**
> >
> > Thanks authors, your response clarified all of my comments and questions. My initial positive assessment remains positive.

---

### Official Review · Reviewer_3D7c · 2021-07-13

**Rating:** 7
**Confidence:** 5

**Summary:**

The paper suggests a new generative model for 3D-aware image synthesis by generating radiance fields with an underlying 3D occupancy representation. This choice of representations allows the generative model to learn surfaces, which are rendered to view consistent images from arbitrary poses using volume rendering. During training, the rendering is performed using samples from a gradually shrinking sampling region around the surface, which encourages the surface to be more compact. The paper demonstrates the method's ability to synthesize images with 3D consistency and further presents their matching shape and normal.

**Limitations And Societal Impact:**

The authors did address the main limitations of the method and its potential social impacts. However, those are mentioned in the supplementary only. I suggest the authors move them or add a short discussion in the main paper on the basics limitation and societal impact. Lastly, I appreciate the author's comment regarding the potential environmental effects.

**Main Review:**

(A) Those are the main strengths and novelty I find in the paper:
1. The 3D surface representation results learned from unposed images are pretty impressive, based on the fact that they are learned in an unsupervised manner.
2. The paper is well written and has a good flow. The related works section presents the method's origin story and binds the reader to the relevant area. Also, in the method section, the authors give the background for the neural radiance field naturally and easily.
3. In the case of representing object surfaces, the motivation for more concentrated volume densities is evident. Therefore, the use of occupancy representations, which could inherently ensure deterministic surfaces, is good and intuitive for the task of generating compact surfaces (like the presented faces and cats).
4. I appreciate the presented measure of the weights' variance, as it indicates the method compactness compared to the other baselines.

(B) Following are the weaknesses I find in the presented method and the concerns which I would like to be addressed by the authors:
1. The need for compact surfaces is clear for objects surfaces (like faces and cats), but it is not entirely trivial for me why the previous works are more prone to predict diffuse surfaces. Namely, why directly training equation 2 (which is the optimization that the basslines GRAF and pi-GAN follows) fails to maintain the surface compactness? The authors suggest that it may arise from the shape-color ambiguity, yet I see this ambiguity as another existing problem for other geometrical defects that can exist even with compact surfaces. Hence, my biggest concern that arises is the method's key contribution compare to its basslines. There are two different aspects of the presented method compared to its baseline: the 3D representation and the sampling procedure. My question is, which one of those is "responsible" for the surface compactness? Is it the sampling procedure around the surface that is shrinking and facilitating the weights' concentration, or is it the underlying surface representation? I would like if the authors can suggest an adequate ablation study to answer this concern or give a better intuition for why the baselines tend to diffuse densities.
2. The authors present the evaluation of the generated images as done in standard GAN's image models, but I find the generated surfaces' evaluation to be insufficient. Since the paper's central claim is better modeling of the 3D representation, the evaluation should support that claim. I find the geometric evaluation to rely on the correctness of the CNN to predict depth maps. Nonetheless, if I understand correctly, the depth maps from the presented method, and the two other baselines are taken as the expected depth, not according to some level set as presented in the figures. A better evaluation would be to show that the generated geometry preserves similar geometry information as in the data set (e.g., mean curvature, mean geodesic distance of random points, faces are and so on). I suggest the authors present other possible evaluations for the underlying geometry or give a better motivation for their proposed evaluation.
3. I believe the authors have an error in equation 9 (the opacity regularization) where each element should be multiplied by its probability, namely: p*log(p)+(1-p)log(1-p). This will give the entropy of the binary probability alpha, and minimizing this regularization aims to drive each sample on the ray to either 1 or 0. If there is no error here, I would like to get an explanation or intuition regarding this formula.
4. Regarding the presented ablation study: it is not clear from the text what is the corresponding experiment for each image in figure 6. Moreover, the ablation should check the contribution of each component (regularization prior) separately. As for what I understand, the authors include results of either with and without both priors. I would appreciate it if the authors can supply an ablation of removing one regularization at a time, to determine the effect of each regularization.

(C) Some other minor questions and suggestions to the authors:
1. I wonder why it seems there is a problem with modeling the eyes geometry (more noticeable in the CelebA). I suspect it relates to the fact the light field is a bit more complicated to present, but I would like to hear the author's opinion.
2. How did the authors extract the shape (shown in the figures) from the pi-GAN and GRAF methods? Namely, how the level set was chosen?
3. Line 150 says that the density varies between -10 to 50, but the physical density should be positive.
4. I suggest the authors present the concrete differences for the other baselines (pi-GAN and GRAF) in the context of the radiance field modeling and sampling procedure. It would help to understand the authors' contribution and novelty compared to existing baselines.

**Time Spent Reviewing:**

8

---

> ### Author Response · Authors · 2021-08-10
> **Feedback to reviewer 3D7c**
>
> **Q1: Which one is “responsible” for surface compactness? The shrinking sampling procedure or the proposed 3D representation?**
>
> **A1**: The combination of both ensures surface compactness. Actually, we get our intuition of 3D representation from its characteristic that occupancy networks encourage compact object surfaces inherently. Meanwhile, the shrinking sampling procedure motivates the model to predict compact object surfaces. Besides, we add the shrinking process into baseline methods like GRAF or pi-GAN and find the problem of diffused surfaces can’t be circumvented, which demonstrates that only the shrinking procedure cannot guarantee a compact object surface.
>
> **Q2: Other possible evaluation protocols for the underlying geometry.**
>
> **A2**: Thanks for your suggestion. As shown in the tables below, we extra report mean curvature (MC) and mean geodesic distance (MGD) on BFM, CelebA and Cats datasets respectively, from which we can see that GOF still outperforms baseline methods.
>
> | Datasets | BFM | | CelebA | | Cats | |
> | :--- | :---: | :---:  | :---:  | :---: | :---:  | :---: |
> | | MC $\downarrow$ | MGD $\downarrow$ | MC $\downarrow$ | MGD $\downarrow$ | MC $\downarrow$ | MGD $\downarrow$ |
> | pi-GAN | 16.84 | 0.483 | 25.94 | 0.450 | 34.05 | 0.494 |
> | GOF   | 12.25 | 0.226 | 23.13 | 0.231 | 30.14 | 0.317 |
>
> These new metrics will be reported in the revision.
>
> **Q3: An explanation for equation 9 (the opacity regularization).**
>
> **A3**: Considering that the occupancy values are in [0, 1], we use Beta distribution B(0.5, 0.5) to model the opacities. In our paper, the negative log-likelihood of the Beta distribution is regarded as the entropy of the opacity, following [1].
>
> [1] Lombardi, Stephen, et al. "Neural volumes: learning dynamic renderable volumes from images." ACM Transactions on Graphics (TOG) 38.4 (2019): 1-14.
>
> **Q4: More detailed ablation study on the regularization priors.**
>
> **A4**: Thanks for pointing it out. As suggested, we remove one regularization at a time and report the quantitative results on BFM, CelebA, and Cats datasets in the table below.
>
> | Datasets | BFM | | CelebA | | Cats | |
> | :--- | :---: | :---: | :---: | :---: | :---: | :---: |
> | | FID $\downarrow$ | $\Sigma_{w_i} \downarrow$ | FID $\downarrow$ | $\Sigma_{w_i} \downarrow$ | FID $\downarrow$ | $\Sigma_{w_i} \downarrow$  |
> | GOF w/o both priors |  16.4 | 1.443 | 17.2 | 4.782 | 19.3 | 0.591 |
> | GOF w/o normal prior | 15.4 | 1.430 | 16.7 | 4.766 | 18.8 | 0.578 |
> | GOF w/o opacity prior | 16.2 | 1.439 | 15.5 | 4.213 | 17.4 | 0.546 |
> | GOF   |  15.3 | 1.426 | 15.0 | 4.115 | 17.1 | 0.533 |
>
> As illustrated in the table, our GOF model on synthetic dataset BFM benefits a lot from the opacity prior. Meanwhile, the normal regularization also boosts the GOF model especially on in-the-wild datasets like CelebA and Cats.
>
> **Q5: The problem of modeling the eyes geometry.**
>
> **A5**: We have also observed this phenomenon in the experiments and come up with two reasons for it. Firstly, the light field of the eyes is more complicated than other regions. More importantly, the dataset is biased (especially in the CelebA), where people always gaze at the camera when taking photos, the biased eye poses are inadequate to provide multi-view information for modeling eyes accurately.
>
> **Q6: How was the level set chosen in the pi-GAN and GRAF methods?**
>
> **A6**: We just take the provided values if they’re available in the paper. Otherwise, we will apply the bisection method to find the most suitable values.
>
> **Q7: The density values should be positive.**
>
> **A7**: Yes. In the paper, we refer to the values before the ReLU activation function. It might be a little bit misleading. We will clarify this in the revision.
>
> **Q8: The concrete differences with baselines.**
>
> **A8**: Thanks for your advice. We will add these concrete differences with baselines in the methodology section.

---

> > ### Comment · Reviewer_3D7c · 2021-08-16
> > **Feedback**
> >
> > I thank the authors for addressing my concerns. I believe the additional experiment and evaluations would emphasize the contribution of the presented method. \
> > I have some follow-up questions regarding some of the answers:
> >
> > Regarding Q1, I wonder if using GOF representation and uniform sampling (rather than the shrinking process) still leads to a compact surface. I don't require the authors to perform this experiment, rather give their intuition for the results based on their experiment. I also suggest they add this experiment in addition to the experiments of running the baselines with the shrinking sampling to show the contribution of the combination.
> >
> > Another question regarding this concern is why/how the
> > "shape-color ambiguity" (described in lines 136-139) is prevented in GOF?
> >
> > Regarding Q3, my concern is more numerical, as I don't understand how this loss is not exploding for alpha values which are exactly 0 or 1.

---

> > > ### Author Response · Authors · 2021-08-16
> > > **Follow-up response to reviewer 3D7c**
> > >
> > > Thanks for your instructive comments and feedback.
> > >
> > > We'll first answer your query about the "shape-color" ambiguity. Baseline methods always sample in a large region, making the training of different rays heavily interfere with each other. This results in considerable shape-color ambiguity. By contrast, our proposed shrinking process for the sampling region can help to mitigate such ambiguity in GOF. Intuitively, using GOF representation alone is less effective in encouraging a compact surface. We will add these ablation experiments in the revision to show the contribution of the combination.
> > >
> > > For your concern on opacity prior in Eq.9, we added 0.1 inside the log term in the implementation to avoid the exploding.
> > > $$ L_\text{opacity} = \frac{1}{N} \sum_{i=1}^{N} \log(0.1 + \alpha_\theta(\mathbf{x}_i, \mathbf{z})) + \log(0.1 + 1 - \alpha_\theta(\mathbf{x}_i, \mathbf{z})). $$

---

> > > > ### Comment · Reviewer_3D7c · 2021-08-16
> > > > **Thanks**
> > > >
> > > > Thank you, this addresses all the remaining concerns I had.

---

### Official Review · Reviewer_mS1W · 2021-07-15

**Rating:** 7
**Confidence:** 4

**Summary:**

The paper proposes GOF, a method to improve the geometry and quality of generative methods that use a neural radiance field representation. The core idea is to modify the network's output to predict occupancy which can be used to find the surface of the rendered object. Once the surface is known, the network can be sampled in a region around the surface that shrinks during training. This accomplishes two goals, it consolidates the geometry to a surface and requires fewer samples during inference. When compared to existing approaches, the depths and normals are cleaner and the image quality metrics are marginally better.

**Limitations And Societal Impact:**

This method is more restrictive in the types of scenes it can represent as it assumes each ray intersects with a single surface. This should be listed in the limitations.


**Main Review:**

I’ve listed my primary concerns below. Overall the paper is well written and the method is simple and straightforward. For these reasons I think this paper is marginally above the acceptance threshold.

a) The paper never actually demonstrates why the recovered representation is better than existing methods (other than showing a depth map). It is noted that downstream tasks like relighting could be improved, but is not demonstrated. A good portion of the paper also discusses the ability to reduce the number of samples taken during inference which could help with “applications on mobile devices”. It would have been nice to see quantitative values demonstrating the speed up of the proposed method. As the paper currently stands, the quality of the image outputs look fairly similar to pi-GAN. I think this paper would be much stronger if advantages on downstream tasks or a speedup compared to baselines were demonstrated.

b) How much can we attribute the improved depth and normals on the use of occupancy compared to the included priors? It would have been nice to see qualitative examples of pi-GAN or GRAF trained with the extra priors. I am also slightly concerned about the quantitative evaluation. Since a different CNN is trained for each baseline, it seems that baselines with larger image FID will be further out of distribution when testing the CNN on the ground truth test images. How do you know that errors in the predicted test depths are a result of bad depths from the baseline versus a distribution mismatch on the CNN input?

c) GOF focuses the evaluation on forward facing faces. It is argued that the smaller camera baseline causes the compared methods to “diffuse” the geometry. GRAF and pi-GAN also test on a 360 scene of synthetic cars and appear to recover fairly good geometry. For scenes with larger baselines is GOF still necessary or do existing approaches work well enough?

d) The conversion of the network to output alpha instead of density forgoes the variable dictating the distance between samples (delta). While the samples are uniform, the distance does change during training because the sampling interval shrinks. Is this something that was considered and is it an issue for this model?

NIT:
LN 177 “a” -> “at”
LN 222 “ms = ? times”

**Time Spent Reviewing:**

4

---

> ### Author Response · Authors · 2021-08-10
> **Feedback to reviewer mS1W**
>
> **Q1: Quantitative values demonstrating the speedup during inference; advantages on downstream tasks like relighting.**
>
> **A1**: As suggested, we estimate the inference speed of both pi-GAN and GOF when rendering 256x256 images using a single CPU. On average, pi-GAN costs about 78s per image, while GOF takes about 56s, saving approximately 28% of the time. In terms of memory usage, when training with 128x128 images, GOF only costs 3.8G GPU memory, compared to 7G of baseline methods, leading to a reduction of 46%. For the relighting task, we employ Unsup3d[35] to predict the albedo for generated images. Baseline methods like GRAF or pi-GAN suffer from diffused object surfaces so that the rendered images under various light conditions contain clear artifacts. By contrast, GOF can always render realistic images under different light conditions. We will add the relighting results in the revision.
>
> **Q2: Qualitative results of GRAF or pi-GAN trained with the extra priors. Which does attribute the improved quantitative results on depth prediction, better shape reconstruction, or just a distribution mismatch?**
>
> **A2**: We have trained GRAF and pi-GAN with the extra priors and still observe the diffused object surfaces and messy normals. For quantitative results, pi-GAN holds the comparable FID score as GOF (15.8 vs 15.0) but underperforms our method clearly on the MAD metric (20.46 vs 14.11). Moreover, we choose an earlier GOF model checkpoint with the same FID score as pi-GAN to evaluate the depth accuracy again. This earlier model obtains 14.89 on the MAD metric and still demonstrates better performance than pi-GAN, which indicates that better shape reconstruction in GOF leads to improved quantitative results on the depth prediction.
>
> **Q3: Is GOF still necessary on other datasets like synthetic cars?**
>
> **A3**: Compared to synthetic datasets, CelebA or Cats is more challenging. As presented in the paper, GOF achieves better performance on these in-the-wild datasets. Meanwhile, we conduct experiments on the synthetic car dataset and find GOF still outperforms the baselines.
>
> | Method | $\Sigma_{w_i} \downarrow$ | MC $\downarrow$ | MGD $\downarrow$ |
> | :--- | :---: | :---: | :---: |
> | pi-GAN | 1.884 | 13.07 | 0.874 |
> | GOF  | 1.507 | 12.39 | 0.831 |
>
> Here, mean curvature (MC) and mean geodesic distance (MGD) are another two evaluation protocols for geometry. The lower, the better. We will add the experiment results to the revision.
>
> **Q4: Is it an issue that the distance between samples (\delta) is ignored?**
>
> **A4**: It’s a good question. For the radiance representation, the volume rendering equation contains the distance \delta, where the predicted radiance values \sigma are inherently distance-dependent. Hence, the distance between samples has to be taken into consideration here. For our proposed occupancy representation, the distance \delta is explicitly omitted as shown in Eq.4. Actually, the shrinking procedure during training is kindly slow and the GOF model will automatically adapt to the varying distance.

---

> > ### Comment · Reviewer_mS1W · 2021-08-12
> > **Rebuttal Feedback**
> >
> > Thank you for addressing my concerns in the review. I appreciate all of the additionally experiments that were run. I think the additional results of lighting comparisons will strengthen the paper.

---

> > > ### Author Response · Authors · 2021-08-13
> > > **Thanks for your comments!**
> > >
> > > Thank you for your constructive review, which helps us improve the quality of the paper.

---

### Decision · Program_Chairs · 2021-09-27

**Decision:**

Accept (Poster)

**Comment:**

The paper received reviews from four expert reviewers in the community. All reviewers agree that the proposed hybrid volume-surface approach is novel in the context of 3D generative modeling. In the initial review, there were some concerns regarding the qualitative evaluation (reviewer mS1W), results on synthetic cars (reviewer mS1W), and evaluation on geometry (reviewer 3D7c), ablation on the regularization prior (reviewer 3D7c). After the rebuttal, the reviewers were convinced by the additional results as well as clarification. The AC reads the reviews, rebuttal, and also agrees that all the concerns raised by the reviewers have been properly addressed with the additional explanation and experimental results. The AC thus recommends to accept and encourages the authors to revise and strengthen the paper by incorporating the suggested changes and additional evaluation.